# MQES: Max-Q Entropy Search for Efficient Exploration in Continuous Reinforcement Learning

## Abstract

The principle of optimism in the face of (aleatoric and epistemic) uncertainty has been utilized to design efficient exploration strategies for Reinforcement Learning (RL). Different from most prior work targeting at discrete action space, we propose a generally information-theoretic exploration principle called Max-Q Entropy Search (MQES) for continuous RL algorithms. MQES formulates the exploration policy to maximize the information about the globally optimal distribution of $Q$ function, which could explore optimistically and avoid over-exploration by recognizing the epistemic and aleatoric uncertainty, respectively. To make MQES practically tractable, we firstly incorporate distributional and ensemble $Q$ function approximations to MQES, which could formulate the epistemic and aleatoric uncertainty accordingly. Then, we introduce a constraint to stabilize the training, and solve the constrained MQES problem to derive the exploration policy in closed form. Empirical evaluations show that MQES outperforms state-of-the-art algorithms on Mujoco environments.

## 1 Introduction

In Reinforcement Learning (RL), one of the fundamental problems is exploration-exploitation dilemma, i.e., the agents explore the states with imperfect knowledge to improve future reward or instead maximize the intermediate reward at the perfectly understood states. The main obstacle of designing efficient exploration strategies is how the agents decide whether the unexplored states leading high cumulative reward or not.

Popular exploration strategies, like $\epsilon$-greedy (Sutton & Barto, 1998) and sampling from stochastic policy (Haarnoja et al., 2018), lead to undirected exploration through additional random permutations. Recently, uncertainty of systems are introduced to guide the exploration (Kirschner & Krause, 2018; Mavrin et al., 2019; Clements et al., 2019; Ciosek et al., 2019). Basically, as Moerland et al. (2017) points out, two source of uncertainty exists in the RL system, i.e., epistemic and aleatoric uncertainty. Epistemic uncertainty is also called parametric uncertainty, which is the ambiguity of models arisen from the imperfect knowledge to the environment, and can be reduced with more data. Aleatoric uncertainty is an intrinsic variation associated with the environment, which is caused by the randomness of environment, and is not affected by the model. In the RL system, if the states are seldom visited, the epistemic uncertainty at these states are relatively large. Hence, the exploration methods should encourage exploration when epistemic uncertainty is large. Moreover, heteroscedastic aleatoric uncertainty means that different states may have difference randomness, which renders different aleatoric uncertainty. If we do not distinguish these two uncertainties and formulate them separately, we may explore the states visited frequently but with high randomness, i.e., low epistemic uncertainty and high aleatoric uncertainty, which is undesirable.

By introducing uncertainty, the exploration objectives like Thompson Sampling (TS) (Thompson, 1933; Osband et al., 2016) and Upper Confidence Bound (UCB) (Auer, 2002; Mavrin et al., 2019; Chen et al., 2017) are utilized to guide the exploration in RL. However, since the aleatoric uncertainty in the RL systems are heteroscedastic, i.e., the aleatoric uncertainty depends on states and actions and can be different, the above methods are not efficient. Hence, Nikolov et al. (2019) proposes novel exploration objective called Information-Directed Sampling (IDS) accounting for

epistemic uncertainty and heteroscedastic aleatoric uncertainty. However, these methods (Nikolov et al., 2019; Mavrin et al., 2019; Chen et al., 2017; Osband et al., 2016) can only be applied in the environment with discrete action space.

In this paper, we propose a generally information-theoretic principle called Max-Q Entropy Search (MQES) for off-policy continuous RL algorithms. Further, as an application example of MQES, we combine distributional RL with soft actor-critic method, where the epistemic and aleatoric uncertainty are formulated accordingly. Then, we incorporate MQES to Distributional Soft Actor-Critic (DSAC) (Ma et al., 2020) method, and show how MQES utilizes both uncertainty to explore. Finally, our results on Mujoco environments show that our method can substantially outperform alternative state-of-the-art algorithms.

## 2 RELATED WORK

Efficient exploration can improve the efficiency and performance of RL algorithms. With the increasing emphasis on exploration efficiency, various exploration methods have been developed. One kind of methods use intrinsic motivation to stimulate agent to explore from different perspectives, such as count-based novelty (Martin et al., 2017; Ostrovski et al., 2017; Bellemare et al., 2016; Tang et al., 2017; Fox et al., 2018), prediction error (Pathak et al., 2017), reachability (Savinov et al., 2019) and information gain on environment dynamics (Houthooft et al., 2016). Some recently proposed methods in DRL, originating from tracking uncertainty, do efficient exploration under the principle of OFU (optimism in the face of uncertainty), such as Thompson Sampling (Thompson, 1933; Osband et al., 2016), IDS (Nikolov et al., 2019; Clements et al., 2019) and other customized methods (Moerland et al., 2017; Pathak et al., 2019).

**Methods for tracking uncertainty.** Bootstrapped DQN (Osband et al., 2016) combines Thompson sampling with value-based algorithms in RL. It is similar to PSRL (Strens, 2000; Osband et al., 2013), and leverages the uncertainty produced by the value estimations for deep exploration. Bootstrapped DQN has become the common baseline for lots of recent works, and also the well-used approach for capturing epistemic uncertainty (Kirschner & Krause, 2018; Ciosek et al., 2019). However, this takes only epistemic uncertainty into account.

Distributional RL approximates the return distribution directly, such as Categorical DQN (C51) (Bellemare et al., 2017), QR-DQN (Dabney et al., 2018b) and IQN (Dabney et al., 2018a). Return distribution can be used to approximate aleatoric uncertainty, but those methods do not take advantage of the return distribution for exploration.

**Exploration with two types of uncertainty.** Traditional OFU methods either focus only on the epistemic uncertainty, or consider the two kinds of uncertainty as a whole, which can easily lead the naive solution to favor actions with higher variances. To address that, Mavrin et al. (2019) studies how to take advantage of distributions learned by distributional RL methods for efficient exploration under both kinds of uncertainty, proposing Decaying Left Truncated Variance (DLTV).

Nikolov et al. (2019) and Clements et al. (2019) propose to use Information Direct Sampling (Kirschner & Krause, 2018) for efficient exploration in RL (IDS for RL), which estimate both kinds of uncertainty and use IDS to make decision for acting with environment. We refer to the practice of uncertainty estimation in (Clements et al., 2019) as shown in Sec. 4.2.1. IDS integrates both uncertainty and has made progress on the issue of exploration, but this is limited on discrete action space. We do focus on how best to exploit both uncertainty for efficient exploration in a continuous action space in our paper.

**Optimistic Actor Critic.** More closely related to our work is the paper of OAC (Ciosek et al., 2019), which uses epistemic uncertainty to build the upper bound of Q estimation $Q^{\text{UB}}$. OAC is based on Soft Actor-Critic (SAC) (Haarnoja et al., 2018), additionally proposing exploration bonus to facilitate exploration. Despite the advantages that OAC has achieved over SAC, it does not consider the potential impact of the aleatoric uncertainty, which may cause misleading for exploration.

## 3 PRELIMINARIES

### 3.1 DISTRIBUTIONAL RL

Distributional RL methods study distributions rather than point estimates, which introduce aleatoric uncertainty from distributional perspective. There are different approaches to represent distribution in RL. In our paper, we focus on quantile regression used in QR-DQN (Dabney et al., 2018b), where the randomness of state-action value is represented by the quantile random variable $Z$ with value $z$. $Z$ maps the state-action pair to a uniform probability distribution supported on $z_i$, where $z_i$ indicates the value of the corresponding quantile estimates. If $\tau_i$ is defined as the quantile fraction, the cumulative probabilities of such quantile distribution is denoted by $F_Z(z_i) = Pr(Z < z_i) = \tau_i = 1/N$ for $i \in 1, ..., N$.

Similar to the Bellman operator in the traditional Q-Learning (Watkins & Dayan, 1992), the distributional Bellman operator $\mathcal{T}_D^\pi$ under policy $\pi$ is given as:

$$\mathcal{T}_D^\pi Z(s_t, a_t) \overset{D}{=} R(s_t, a_t) + \gamma Z(s_{t+1}, a_{t+1}), \quad a_{t+1} \sim \pi(\cdot|s_{t+1}). \tag{1}$$

Notice that this operates on random variables, $\overset{D}{=}$ denotes that distributions on both sides have equal probability laws. Based on the distributional Bellman operator, Dabney et al. (2018b) proposes QR-DQN to train quantile estimations via the quantile regression loss, which is denoted as:

$$\mathcal{L}_{QR}(\theta) = \frac{1}{N} \sum_{i=1}^{N} \sum_{j=1}^{N} [\rho_{\hat{\tau}_i}(\delta_{i,j})] \tag{2}$$

where $\delta_{i,j} = R(s_t, a_t) + \gamma z_j(s_{t+1}, a_{t+1}; \theta) - z_i(s_t, a_t; \theta)$, $\rho_\tau(u) = u * (\tau - \mathbf{1}_{u<0})$, and $\hat{\tau}_i$ means the quantile midpoints, which is defined as $\hat{\tau}_i = \frac{\tau_{i+1} + \tau_i}{2}$.

### 3.2 DISTRIBUTIONAL SOFT ACTOR-CRITIC METHODS

Following Ma et al. (2020), Distributional RL has been successfully integrated with soft Actor-Critic (SAC) algorithm. Here, considering the maximum entropy RL, the distributional soft Bellman operator $\mathcal{T}_{DS}^\pi$ is defined as follows:

$$\mathcal{T}_{DS}^\pi Z(s_t, a_t) \overset{D}{=} R(s_t, a_t) + \gamma[Z(s_{t+1}, a_{t+1}) - \alpha \log \pi(a_{t+1}|s_{t+1})] \tag{3}$$

where $a_{t+1} \sim \pi(\cdot|s_{t+1}), s_{t+1} \sim \mathcal{P}(\cdot|s_t, a_t)$. The quantile regression loss in DSAC is different from original QR-DQN only on the $\delta_{i,j}$ by considering the maximum entropy RL framework. DSAC extends the clipped double Q-Learning proposed on TD3 (Fujimoto et al., 2018) to overcome the overestimation problem. Two Quantile Regression Deep Q Networks have the same structure that are parameterized by $\theta_k, k = 1, 2$. Following the clipped double Q-Learning, the TD-error of DSAC is defined as:

$$y_i^t = \min_{k=1,2} z_i(s_{t+1}, a_{t+1}; \bar{\theta}_k) \tag{4}$$

$$\delta_{i,j}^k = R(s_t, a_t) + \gamma[y_i^t - \alpha \log \pi(a_{t+1}|s_{t+1}; \bar{\phi})] - z_j(s_t, a_t; \theta_k) \tag{5}$$

where $\bar{\theta}$ and $\bar{\phi}$ represents their target networks respectively. DSAC has the modified version of critic, while the update of actors is unaffected. It is worth noticing that the state-action value is the minimum value of the expectation on certain distributions, as

$$Q(s_t, a_t; \theta) = \min_{k=1,2} Q(s_t, a_t; \theta_k)$$

$$= \frac{1}{N} \min_{k=1,2} \sum_{i=0}^{N-1} z_i(s_t, a_t; \theta_k) \tag{6}$$

Thus, in DSAC, the original problem aims to maximize the following objective function:

$$\mathcal{J}_\pi(\phi) = \mathbb{E}_{s_t \sim \mathcal{D}, \epsilon \sim \mathcal{N}}[\log \pi(f(s_t, \epsilon_t; \phi)|s_t) - Q(s_t, f(s_t, \epsilon_t; \phi); \theta)], \tag{7}$$

where $\mathcal{D}$ is the replay buffer, $f(s_t, \epsilon_t; \phi)$ means sampling action with re-parameterized policy.

## 4 ALGORITHM

This paper proposes a new exploration principle for continuous RL algorithms, i.e., MQES, which leverages epistemic and aleatoric uncertainties to explore optimistically and avoid over-exploration. To make MQES practically tractable, distributional and ensemble $Q$ function approximations are introduced to formulate the epistemic and aleatoric uncertainty accordingly. Nevertheless, a constraint is introduced in the MQES to stabilize the training, and the approximated exploration policy is derived in the closed form. All these mechanisms are detailed in the following sections accordingly.

### 4.1 EXPLORATION STRATEGY: MAX-Q ENTROPY SEARCH

To achieve a better exploration, MQES derives an exploration policy $\pi_E$ which aims at reducing the epistemic uncertainty and obtain more knowledge of the globally optimal $Q$ function.

Firstly, we define exploration action random variable $A_E \sim \pi_E(a|s)$ with value $a_E \in \mathbf{A}$, where $\mathbf{A}$ is the action space. $Z^*(s, a^*)$ is the random variable following the distribution describing the randomness of return obtained by globally optimal policy $\pi^*$, and the value of $Z^*(s, a^*)$ is defined as $z^*(s, a^*)$. Through maximizing the mutual information between random variables $Z^*(s, a^*)$ and $A_E$, we reduce the uncertainty of globally optimal $Q$ function $Q^*$ to encourage exploration.

Specifically, at timestamp $t$, we find the exploration policy $\pi_E$ in the candidate distribution family $\Pi$ that can maximize the information about the optimal action random variable $A^*$ as follows:

$$\pi_E = \arg\max_{\pi \in \Pi} \mathbf{F}^\pi(s_t), \tag{8}$$

where $\mathbf{F}(\cdot)$ is the mutual information and can be written as follows:

$$\begin{aligned} \mathbf{F}^\pi(s_t) &= \mathbf{MI}(Z^*(s, a^*), A|s = s_t) \\ &= \mathbf{H}\left[\pi(a_t|s_t)\right] - \mathbf{H}\left[p(a_t|z^*(s_t, a^*), s_t)\right]. \end{aligned} \tag{9}$$

Here $\mathbf{MI}(\cdot)$ and $\mathbf{H}(\cdot)$ denote the mutual information and entropy of the random variable, respectively. To obtain exploration policy $\pi_E$, we need to measure the posterior probability $p(\cdot)$ in the above equation. For simplicity, we omit the timestamp $t$ in the following.

To measure the posterior probability $p(a|a^*, s)$, we propose the following proposition.

**Proposition 1.** *Generally, the posterior probability is estimated as follows:*

$$p(a|z^*(s, a^*), s) \propto \pi(a|s)\Phi_{Z^\pi(s,a)}(z^*(s, a^*)), \tag{10}$$

*where $\Phi_x$ is the cumulative distribution function (CDF) of $x$, $Z^*$ and $Z^{\pi_E}$ are the random variables, whose distributions describing the randomness of the returns obtained by optimal policy $\pi^*$ and exploration policy $\pi_E$, respectively, and $z^*$ is the value of random variable $Z^*$. (see proof in Appendix A).*

To measure the intractable distribution of $Z^*$ during training, we use the $\hat{Z}^*$ for approximation (i.e., $\hat{Z}^* \approx Z^*$), which will be defined later. In general, $\hat{Z}^*$ is referred to as the optimistic approximator (Mavrin et al., 2019; Chen et al., 2017), and can be formulated using the uncertainty, which will be detailed in Sec. 4.2.1.

Therefore, the $\mathbf{F}^\pi(s)$ in Eq. 9 can be estimated as follows:

$$\mathbf{F}^\pi(s) \approx \hat{\mathbf{F}}^\pi(s) = \mathbb{E}_{\pi, \hat{Z}^*}\left[\log \pi(a|s)(G(s, a) - 1) + G(s, a) \log G(s, a)\right], \tag{11}$$

where $G(s, a) = \frac{1}{C} * \Phi_{Z^\pi(s,a)}(\hat{z}^*(s, a))$. Specifically, $G(s, a)$ measures the difference between $Z^\pi$ and $\hat{Z}^*$, i.e., large value of CDF means that $\hat{z}^*$ is much bigger than the mean of $Z^\pi$.

By introducing distributional value functions in Eq. 10 to estimate the posterior probability, we can use the uncertainty of value function to encourage the exploration, which will be discuss in Sec. 4.2.2.

### 4.2 MQES-BASED EXPLORATION FOR MODERN RL ALGORITHMS

In this section, we propose a scheme to incorporate exploration policy derived from MQES to existing policy-based algorithms, e.g., SAC and TD3, which renders the stable and well-performed algorithm with more efficient exploration.

First, to obtain exploration policy, we employ a constraint to ensure the difference between the exploration and target policies within a certain range (i.e., $\mathbf{KL}(\pi||\pi_T) \leq \alpha$). The target policy $\pi_T$ here, we mean the policy learned by any existing policy-based algorithms. It is worth noting that MQES introduces distributional and ensemble critics to the existing framework (e.g., introducing distributional critic to SAC formulates DSAC). Moreover, we utilize the critic of target policy to formulate $\hat{Z}^*$ and $Z^{\pi_E}$, which will be stated later.

Intuitively, introducing the constraint in MQES ensures that the critic of target policy guides the exploration properly and stabilizing the training. Otherwise, the exploration could be ineffective, and the update of target policy can be dramatically bad. Specifically, if the difference between $\pi$ and $\pi_T$ are with significant difference, $Z^{\pi_T}$ could not criticize exploration policy properly, and it may explore with wrong guidance, where the experiences stored in the replay buffer are with poor quality and the update of target policy fails sequentially.

Second, after introducing the KL constraint, the MQES-based exploration for modern RL algorithms is given as follows:

$$\pi_E(a|s) = \arg\max_{\pi} \quad \hat{\mathbf{F}}^{\pi}(s),$$
$$s.t. \quad \mathbf{KL}(\pi||\pi_T) \leq \alpha, \tag{12}$$

where both the exploration $\pi_E = \mathcal{N}(\mu_E, \Sigma_E)$ and target policy $\pi_T = \mathcal{N}(\mu_T, \Sigma_T)$ are Gaussian distributions. By expanding $\hat{\mathbf{F}}^{\pi}$ linearly, we solve the problem in Eq. 12 using the following proposition:

**Proposition 2.** *The MQES exploration policy $\pi_E = \mathcal{N}(\mu_E, \Sigma_E)$ derived from Eq. 12 is as follows:*

$$\mu_E = \mu_T + \frac{\sqrt{2\alpha}}{\left\| \mathbb{E}_{\hat{Z}^*} \left[ m \odot \frac{\partial \hat{Z}^*(s,a)}{\partial a}|_{a=\mu_T} \right] \right\|_{\Sigma_E}} \Sigma_E \mathbb{E}_{\hat{Z}^*} \left[ m \odot \frac{\partial \hat{Z}^*(s,a)}{\partial a}|_{a=\mu_T} \right], \Sigma_E = \Sigma_T. \tag{13}$$

*In specific, the $i$-th element of vector $m$ is $m_i = \log \frac{G(s,\mu_T)}{\sqrt{(2\pi)\Sigma_{ii}}} + 1$, $G(s, \mu_T) = \frac{\Phi_{Z^{\pi_E}(s,\mu_T)}(\hat{Z}^*(s,\mu_T))}{C}$, $i \in \{1,...,n\}$ and $n$ is the action dimension (see proof in Appendix B).*

It is worth noting that the expectation against $\mathbb{E}_{\hat{Z}^*}$ can be estimated by sampling, and the estimation of Eq. 13 is as follows:

$$\mu_E = \mu_T + \frac{\sqrt{2\alpha}}{K} \sum_{i=1}^{K} \frac{1}{\left\| m \odot \frac{\partial \hat{Z}_i^*(s,a)}{\partial a}|_{a=\mu_T} \right\|_{\Sigma_E}} \Sigma_E \left[ m \odot \frac{\partial \hat{Z}_i^*(s,a)}{\partial a}|_{a=\mu_T} \right], \tag{14}$$

### 4.2.1 FORMULATION OF $\hat{Z}^*$ AND $Z^{\pi_E}$

In this section, we formulate the epistemic and aleatoric uncertainty with the critic of target policy, thereby distributions of $\hat{Z}^*$ and $Z^{\pi_E}$ can be estimated. The remaining parts describe how to achieve these two estimations, respectively.

**Formulation of $\hat{Z}^*$.** In order to formulate the distribution of estimated optimal $Q$ value, i.e., $\hat{Z}^*$, we firstly estimate its upper confidential bound, denoted by $Q^{\mathrm{UB}}$. Aligned with Clements et al. (2019), we adopt two independent distribution approximators $Z(s, a; \theta_1)$ and $Z(s, a; \theta_2)$ parameterized by $\theta_1$ and $\theta_2$, respectively. Then we measure the epistemic uncertainty first as follows:

$$\sigma_{\mathrm{epistemic}}(s, a; \theta) = \frac{1}{2} \mathbb{E}_{i \sim \mathcal{U}(1,N)} |z_i(s, a; \theta_1) - z_i(s, a; \theta_2)|, \tag{15}$$

where $N$ is the number of quantiles, and $z_i(s, a; \theta)$ is the value of the $i$-th quantile drawn from $Z(s, a; \theta)$.

Consequently, the upper confidential bound of $Q$-value is given leveraging the $\sigma_{\text{epistemic}}$ as follows:
$$Q^{\text{UB}}(s, a; \theta) = \mu_Z(s, a; \theta) + \beta \sigma_{\text{epistemic}}(s, a; \theta), \tag{16}$$
where $\mu_Z(s, a; \theta) = \frac{1}{N} \Sigma_{i=1}^N \frac{1}{2} \Sigma_{k=1,2} z_i(s, a; \theta_k)$ is the mean estimation over quantile distributions, $\beta$ determines the magnitude of uncertainty we use. $Q^{\text{UB}}$ is commonly considered as a approximation of the optimal $Q$ value in the existing work (Ciosek et al., 2019; Kirschner & Krause, 2018).

Moreover, as shown in (Dabney et al., 2018b), the aleatoric uncertainty can be captured by return distribution, which can be derived in our method by considering those two quantile distributions as follows:
$$\sigma_{\text{aleatoric}}^2(s, a; \theta) = \text{var}_{i \sim \mathcal{U}(1,N)}[\mathbb{E}_{\theta_k} z_i(s, a; \theta_k)]. \tag{17}$$
Inspired by (Wang & Jegelka, 2017), we adopt $Q^{\text{UB}}$ and $\sigma_{\text{aleatoric}}^2$ as mean and variation and formulate the Gaussian distribution $\hat{Z}^*$ as follows:
$$\hat{Z}^*(s, a; \theta) \sim \mathcal{N}(Q^{\text{UB}}(s, a; \theta), \sigma_{\text{aleatoric}}^2(s, a; \theta)) \mathbf{1}_{\hat{z}^* \geq Q^{\text{UB}}}, \tag{18}$$
where $\hat{Z}^*$ follows truncated Gaussian distribution ensuring the globally optimal constraint, i.e., $\mathbb{E}[Z^*] = Q^* \geq Q^{UB}$. Nevertheless, since the distributions of $Q$ functions describe the aleatoric uncertainty, we set the variance of $\hat{Z}^*$ as the aleatoric uncertainty obtained from Eq. 17.

**Formulation of $Z^{\pi_E}$.** Since the target for critic in the advanced algorithms, like SAC and TD3, is usually estimated pessimistically, we take the pessimistic estimation for $Z^{\pi_E}$ to make MQES compatible with the existing modern RL algorithms. Here we present two modeling approaches: Gaussian and quantile distributions.

Intuitively, we assume $Z^{\pi_E}$ to be the Gaussian distribution with pessimistic estimation as the mean:
$$Z^{\pi_E}(s, a; \theta) \sim \mathcal{N}(Q^{\text{LB}}(s, a; \theta), \sigma_{\text{aleatoric}}^2(s, a; \theta)), \tag{19}$$
where $Q^{\text{LB}}(s, a; \theta) = \mu_Z(s, a; \theta) - \beta \sigma_{\text{epistemic}}(s, a; \theta)$, estimates its lower confidential bound.

On the other hand, as the quantile distribution is a value distribution which naturally formulates the underlying aleatoric uncertainty, we can utilize the quantile functions to model the pessimistic quantile distribution directly, breaking the Gaussian assumption above. Specifically, we take the smaller estimates at each quantile, and then the distribution $Z^{\pi_E}(s, a; \theta)$ is a uniform distribution over each quantile value $z_i^{\pi_E}(s, a; \theta)$, shch as
$$z_i^{\pi_E}(s, a; \theta) = \min_{k=1,2} z_i(s, a; \theta_k). \tag{20}$$

Different from the uni-modal Gaussian distribution, the quantile function is able to represent multi-modal distributions, which is more flexible. As the quantile function represents the inverse function of CDF, meaning that we can easily get a general idea of the properties of this pessimistic quantile distribution. It is worth noting that we can utilize other methods to formulate $Z^{\pi_E}$, like mean estimation, i.e., $\mathbb{E}[Z^{\pi_E}] = \mu_Z(s, a; \theta)$ and $z_i^{\pi_E}(s, a; \theta) = \mathbb{E}_{k=1,2}[z_i(s, a; \theta_k)]$. But, it only affects the choice of hyper-parameter $\beta$ and do not affect the final performance.

---

**Algorithm 1** Exploration policy derived from MQES

---

**Initialise:** Current state $s_t$, current value distribution estimators $\theta_k, k = 1, 2$, current policy network $\phi$, target policy $\pi_T(\cdot|s_t; \phi) \sim \mathcal{N}(\mu_T(s_t; \phi), \sigma_T(s_t; \phi))$
**Output:** the MQES exploration policy $\pi_E$
1: Calculate $Z(s_t, a_t; \theta_k), k = 1, 2$
2: Calculate epistemic uncertainty $\sigma_{\text{epistemic}}(s_t, a_t)$ according to Eq. 15
3: Calculate upper bound $Q^{\text{UB}}(s_t, a_t)$ using $\sigma_{\text{epistemic}}(s_t, a_t)$ according to Eq. 16
4: Calculate aleatoric uncertainty $\sigma_{\text{aleatoric}}^2(s_t, a_t)$ according to Eq. 17
5: Construct $\hat{Z}^*(s_t, a_t)$ using $Q^{\text{UB}}(s_t, a_t)$ and $\sigma_{\text{aleatoric}}^2(s_t, a_t)$ (see Eq. 18)
6: Construct $Z^{\pi_E}(s_t, a_t)$ according to Eq. 19 / 20
7: Calculate $\mu_E$ using $Z^{\pi}(s_t, a_t)$ and $\hat{Z}^*(s_t, a_t)$ according to Eq. 14
8: return $\pi_E \sim \mathcal{N}(\mu_E, \sigma_T(s_t; \phi))$

---

The above Alg. 1 summarizes the overall procedure of MQES, including the estimation of uncertainty (Line 2, 4) and the upper confidential bound $Q^{\text{UB}}$ (Line 3), formulation of $Z^{\pi_E}$ and $\hat{Z}^*$ (Line 5-6) and exploration policy generation using KL constraint (Line 7). The generated exploration policy can be adopted by any modern policy-based RL algorithms for an more effective exploration.

### 4.2.2 Analysis of MQES-based Exploration

This section analytically explains how MQES encourages exploration accounting for the aleatoric and epistemic uncertainty. For simplicity, we assume that the sample number is $K = 1$, and Eq. 14 degrades to:

$$\mu_E = \mu_T + \frac{\sqrt{2\alpha}}{\left\| m \odot \frac{\partial \hat{z}^*(s,a)}{\partial a} \big|_{a=\mu_T} \right\|_{\Sigma_E}} \Sigma_E \left[ m \odot \frac{\partial \hat{z}^*(s,a)}{\partial a} \big|_{a=\mu_T} \right], \tag{21}$$

where $m_i = \log \frac{G(s, \mu_T)}{\sqrt{(2\pi_E)\Sigma_{ii}}} + 1$.

Take Gaussian MQES for example, we have $Z^{\pi_E}(s,a) \sim \mathcal{N}(Q^{LB}(s,a), \sigma^2_{\text{aleatoric}}(s,a))$, and the bias term added to $\mu_E$ is decided by the epistemic and aleatoric uncertainty.

Specifically, since epistemic uncertainty is involved in $\hat{Z}^*(s,a)$, the gradient $\frac{\partial \hat{z}^*(s,a)}{\partial a}$ encourages the optimistic exploration. Epistemic uncertainty-based exploration, can avoid the pessimistic underexploration. The aleatoric uncertainty is introduced by CDF $\Phi$. If we have two state-action pairs, i.e., $(s_1, a_1)$ and $(s_2, a_2)$, and $Z^{\pi_E}(s_i, a_i) \sim \mathcal{N}(Q^{LB}(s_i, a_i), \sigma^2_{\text{aleatoric}}(s_i, a_i)), i = 1, 2$, and we assume that $Q^{LB}(s_1, a_1) = Q^{LB}(s_2, a_2)$, $\sigma_1^2 > \sigma_2^2$, and $\hat{z}^*(s_1, a_1) = \hat{z}^*(s_2, a_2)$. Obviously, $\Phi_{Z^\pi(s_1, a_1)}(\hat{z}^*(s_1, a_1)) < \Phi_{Z^\pi(s_2, a_2)}(\hat{z}^*(s_2, a_2))$, which means that larger aleatoric uncertainty leads to smaller action bias.

Therefore, the MQES encourages the exploration by selecting the action increasing the optimistic value function, and avoid the over-exploration by setting smaller action bias at the state, where the aleatoric uncertainty is high.

## 5 Experiments

MQES is designed for efficient exploration in continuous action space problem in RL, allowing the agent to be aware of explore directions that may lead to higher optimistic value function with smaller aleatoric uncertainty. Comparisons between MQES and state-of-the-art algorithms are conducted to verify the MQES regarding the effectiveness and efficiency. Empirical evaluations show that MQES outperforms state-of-the-art algorithms on a series of continue control tasks.

### 5.1 Implementation Details and Experiment Settings

We compare MQES against SAC (Haarnoja et al., 2018) and its distributional variant DSAC (Ma et al., 2020). Ma et al. (2020) also shows the performance of TD4, which is the distributional extension of TD3 (Fujimoto et al., 2018), and can also be used to capture epistemic and aleatoric uncertainty as is pointed out in Sec. 4.2. However, DSAC outperforms TD4 as shown in Ma et al. (2020), so we evaluate based only on SAC and DSAC, and further implement MQES based on DSAC in order to develop the exploration ability.

The training process of MQES is the same as in DSAC, except for the behavior policy used, while we enrich the experience replay with the data generated by $\pi_E$. The pseudo code of the whole process can be found in Appendix C. In order to ensure a fair comparison, the hyper-parameters of DSAC and MQES are the same (see Appendix D). In addition, we have 3 hyper-parameters associated with MQES. The parameter $\sqrt{2\alpha}$ controls the exploration level, and $\beta$ determines the magnitude of uncertainty we use, and $C$ is the normalization factor.

We implement both approaches for building $Z^\pi$ as illustrated in Sec. 4.2.1, and we use MQES_G and MQES_Q to indicate respectively to Gaussian distribution and quantile distribution. We test MQES on several tasks in Mujoco (Todorov et al., 2012), including standard version, as well as modified sparse version and stochastic version. We limit the maximum length of each episode to 100. We run 1250 or more epochs for each task where there are 100 training steps per epoch, with evaluating every epoch where each evaluation reports the average undiscounted return with no exploration noise.

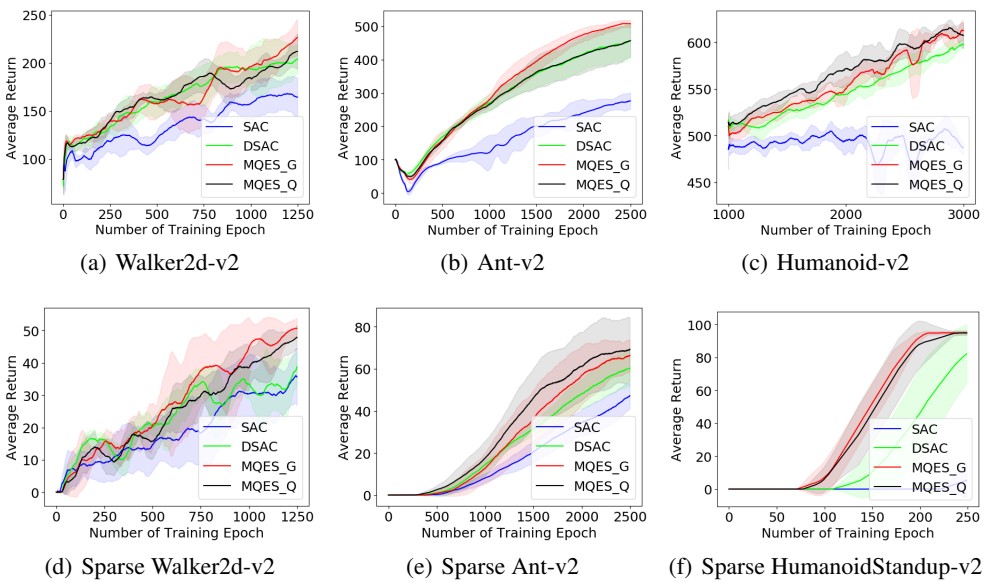

Figure 1: Training curves on continuous control benchmarks in Mujoco. The x-axis indicates number of training epoch (100 environment steps for each training epoch), while the y-axis is the evaluation result represented by average episode return. The shaded region denotes one standard deviation of average evaluation over 5 seeds. Curves are smoothed uniformly for visual clarity.

## 5.2 STANDARD MUJOCO TASKS

We evaluate both MQES_G and MQES_Q in 5 tasks of standard Mujoco, and the result in Figure 1 and Appendix F show that our methods outperform SAC for all those tasks, and also reach better performance than DSAC.

**Performance**. Our results demonstrate that in complex tasks, such as Humanoid-v2 and Ant-v2, our MQES-based exploration policy performs better, while DSAC suffers from the inefficiency caused by deficient exploration. In Ant-v2, DSAC was overtaken in the early stages of training and then MQES stays ahead. Also in Humanoid-v2, the performance of our algorithm always maintains better than DSAC. In some relatively easier tasks, it seems that those tasks are not very demanding for exploration, but MQES performs still at a very advanced level. In Walker2d-v2, MQES and DSAC alternated lead until 1000 epochs, after which MQES had a significant improvement. The final results are shown more specifically in the Table 3.

**Gaussian and Quantile $Z^\pi$.** One can find that, expect for Humanoid-v2, there is no absolute superiority between the two modeling approaches. We hypothesis this is because environment of Humanoid-v2 is more complex than others, and more flexible quantile $Z^\pi$ is needed, which could model the environment more accurately.

## 5.3 SPARSE MUJOCO TASKS

To further show the strength of our algorithm regarding exploration, we evaluate on the sparse version of Mujoco tasks. Specifically, the reward is +1 when the move-distance threshold [1] is reached, otherwise 0. Obviously, since we set the maximum length of episode to 100, the maximum average episode return in the sparse tasks is up to 100.

As shown in Figure 1 and Appendix F, SAC performs poorly in those tasks, which is to be expected, since solving the sparse reward problem requires not only more accurate estimates of critic, but also more efficient exploration. Even in sparse HumanoidStandup-v2 task, when our algorithm MQES

---

[1]We set the threshold according to the statistical analysis of untrained interaction behavior, using 99.9% quantile value, and the threshold of all environments are shown in Appendix E

reaches nearly maximum scores, SAC learns almost nothing, and DSAC performs obviously a bit inferior. As can be seen in combination with the Figure 1 and Table 3, both MQES algorithms perform the best, and both do achieve significantly better results faster than DSAC.

MQES performs extremely well in these sparse environments, which on the one hand demonstrates the importance of exploration when solving sparse reward problems, and on the other hand shows the advantages and capabilities of MQES for efficient exploration, presenting that incorporating uncertainty to exploration could render better performance.

### 5.4 ABLATION STUDY

In this section, we conduct two ablation experiments to show the performance gain by distinguishing aleatoric and epistemic uncertaity and the sensitivity to the hyper-parameters.

#### 5.4.1 GAIN OF ALEATORIC UNCERTAINTY

The exploration with epistemic uncertainty is proved to be efficient (Ciosek et al., 2019) by avoiding the under-exploration. Hence, we present the gain brought by the aleatoric uncertainty by conducting ablation study in this section. Generally, MQES-based actor-critic algorithm degrades to Optimistic Actor-Critic (OAC) (Ciosek et al., 2019) if $m_i = 1$ in Eq. 13. Consequentially, we could compare MQES with OAC to show the necessary of introducing the aleatoric uncertainty to exploration.

In order to show that MQES performs robustly in the face of aleatoric uncertainty, we modified the standard Ant-v2 task by adding heterogeneous noise to the observation to increase the aleatoric uncertainty of the environment. We also extend the OAC to the distributional form (DOAC, Distributional OAC), i.e., to estimate the values in the same way as MQES, in order to ensure a fair comparison. As shown in Figure 2, DOAC does not take into account the aleatoric uncertainty, resulting in its performance being close to that of DSAC. The experimental results confirm that MQES is effective in avoiding the interference of aleatoric uncertainty in the environment and ensuring exploration efficiency.

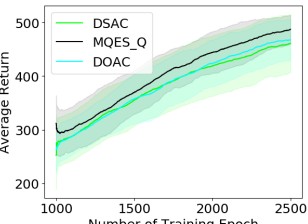

Figure 2: Gain of aleatoric uncertainty, keeping the same plotting settings as Figure 1.

#### 5.4.2 ABLATION STUDY ON HYPER-PARAMETERS

MQES is sensitive to $\sqrt{2\alpha}$, which controls the distance between the behavior policy $\pi_E$ and the target policy $\pi_T$. If $\sqrt{2\alpha}$ is quite small, then the MQES degenerates to DSAC and shows little exploration, and if $\sqrt{2\alpha}$ is larger, then the performance becomes worse because of the gap between behavior policy and target policy. We evaluate the sensitivity to hyper-parameters on Ant-v2 task using MQES_G, and the sensitivity to the constraint $\sqrt{2\alpha}$ is shown in Figure 3, and the sensitivity to $\beta$ is shown in Appendix F, where the error bar indicates half standard deviation.

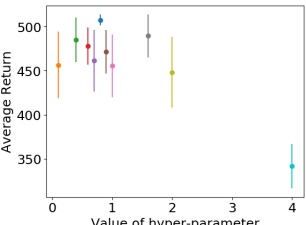

Figure 3: Ablation study on $\sqrt{2\alpha}$.

## 6 CONCLUSION

In this paper, we propose MQES, a generally exploration principle for continuous RL algorithms, which formulates the exploration policy to maximize the information about the globally optimal distribution of $Q$ function. To make MQES practically tractable, we firstly incorporate distributional and ensemble $Q$ function approximations to MQES, which could formulate the epistemic and aleatoric uncertainty accordingly. Secondly, we introduce a constraint to stabilize the training, and solve the constrained MQES problem to derive the exploration policy in closed form. Then, we analyze and show that it explores optimistically and avoid over-exploration by recognizing the epistemic and aleatoric uncertainty, respectively. Empirical evaluations show that MQES works better at the complex environments, where the exploration is needed.

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

## A   PROOF OF PROPOSITION 1

*Proof.* The proof is similar to Wang & Jegelka (2017). Here, we utilize $Z^\pi$ to criticize $\pi$, i.e.,

$$p(a|z^*(s,a^*),s) = \frac{\pi(a|s)\mathbb{E}_{z^\pi \sim Z^\pi}\left[\mathbf{1}_{z^\pi(s,a) \leq z^*(s,a^*)}\right]}{C},$$

where $C$ is the normalization factor, and $\mathbb{E}_{Z^\pi(s,a)}\left[\mathbf{1}_{z^\pi(s,a) \leq z^*(s,a^*)}\right] = \Phi_{Z^\pi(s,a)}(z^*(s,a^*))$    □

## B   PROOF OF PROPOSITION 2

*Proof.* Inspired by the methods of multipliers, it is equivalent to maximize objective function $\hat{\mathbf{F}}^\pi(s,a)$ and minimize constraint $\mathbf{KL}(\pi||\pi_T)$ simultaneously. Firstly, we minimize the $\mathbf{KL}(\pi||\pi_T) = \mathbf{KL}(\mathcal{N}(\mu,\Sigma)||\mathcal{N}(\mu_T,\Sigma_T))$ against the variance matrix $\Sigma$, i.e.,

$$\min_{\Sigma} \mathbf{KL}(\mathcal{N}(\mu,\Sigma)||\mathcal{N}(\mu_T,\Sigma_T)). \tag{22}$$

The optimal solution for problem (22) is

$$\Sigma_E = \Sigma_T. \tag{23}$$

Then, $\hat{\mathbf{F}}^\pi(s,a)$ is expanded linearly around $a_t = \mu_T$:

$$\hat{\mathbf{F}}^\pi(s,a) \approx a^T \nabla_a \hat{\mathbf{F}}^\pi(s,a)|_{a=\mu_T} + const \tag{24}$$

$$= a^T \left[\hat{m} \odot \frac{\partial \hat{z}^*(s,a)}{\partial a}|_{a=\mu_T}\right] + const, \tag{25}$$

where the element of vector $\hat{m} = \{\hat{m}_i\}_{i=1}^n$ is

$$\hat{m}_i = \frac{1}{C}\phi_{Z^\pi(s,\mu_T)}(\hat{z}^*(s,\mu_T))(\log \frac{\Phi_{Z^\pi(s,\mu_T)}(\hat{z}^*(s,\mu_T))}{C\sqrt{(2\pi)\Sigma_{ii}}} + 1), \tag{26}$$

$n$ is the dimension of action, and $\phi(x)$ is the probability distribution function (pdf). Then, problem (12) is reformulated as:

$$\max_{\mu} \quad \mathbb{E}_{\hat{Z}^*}\left\{\mu^T\left[m \odot \frac{\partial \hat{z}^*(s,a)}{\partial a}|_{a=\mu_T}\right]\right\}$$

$$s.t. \quad \frac{1}{2}(\mu-\mu_T)^T\Sigma^{-1}(\mu-\mu_T) \leq \alpha, \tag{27}$$

Then, the Lagrange function of problem (27) is given as:

$$L(\mu) = \mathbb{E}_{\hat{z}^*}\left\{\mu^T\left[\hat{m} \odot \frac{\partial \hat{z}^*(s,a)}{\partial a}|_{a=\mu_T}\right]\right\} - \lambda\left[\frac{1}{2}(\mu-\mu_T)^T\Sigma^{-1}(\mu-\mu_T) - \alpha\right]. \tag{28}$$

According to the KKT condition, we derive the following equations: First, from the stationary, i.e., $\nabla_\mu L(\mu) = m \odot \frac{\partial \hat{Z}^*(s,a)}{\partial a}|_{a=\mu_T} - \lambda\Sigma^{-1}(\mu-\mu_T) = 0$, we get

$$\mu = \mu_T + \frac{1}{\lambda}\Sigma\mathbb{E}_{\hat{Z}^*}\left[\hat{m} \odot \frac{\partial \hat{z}^*(s,a)}{\partial a}|_{a=\mu_T}\right], \tag{29}$$

where $\lambda > 0$. Secondly, since $\lambda > 0$, the constraint is active, i.e., $(\mu-\mu_T)^T\Sigma^{-1}(\mu-\mu_T) = 2\alpha$. Together with Eq. 29, we get the following equation:

$$\lambda = \frac{\left\|\mathbb{E}_{\hat{z}^*}\left[\hat{m} \odot \frac{\partial \hat{Z}^*(s,a)}{\partial a}|_{a=\mu_T}\right]\right\|_\Sigma}{\sqrt{2\alpha}}. \tag{30}$$

Finally, (13) is obtained by plugging (30) to (29).    □

## C   ALGORITHM 2: MQES FOR DSAC

In this section, we show the whole algorithm of our implementation of MQES based on DSAC in Alg. 2.

---

**Algorithm 2** MQES for DSAC

---

**Initialise:** Value networks $\theta_1$, $\theta_2$, policy network $\phi$ and their target networks $\bar{\theta}_1$, $\bar{\theta}_2$, $\bar{\phi}$, quantiles number N, target smoothing coefficient ($\tau$), discount ($\gamma$), an empty replay pool $\mathcal{D}$

1: **for** each iteration **do**
2:      **for** each environmental step **do**
3:          $a_t \sim \pi_E(a_t, s_t)$ according to Alg. 1
4:          $\mathcal{D} \leftarrow \mathcal{D} \cup \{(s_t, a_t, r(s_t, a_t), s_{t+1})\}$
5:      **end for**
6:      **for** each training step **do**
7:          **for** i = 1 to N **do**
8:              **for** j = 1 to N **do**
9:                  calculate $\delta_{i,j}^k$, $k = 1, 2$, following Eq. 5
10:              **end for**
11:          **end for**
12:          Calculate $\mathcal{L}_{QR}(\theta_k)$, $k = 1, 2$ using $\delta_{i,j}^k$ following Eq. 2
13:          Update $\theta_k$ with $\nabla \mathcal{L}_{QR}(\theta_k)$
14:          Calculate $\mathcal{J}_\pi(\phi)$, following Eq. 7
15:          Update $\phi$ with $\nabla \mathcal{J}_\pi(\phi)$
16:      **end for**
17:      Update target value network with $\bar{\theta}_k \leftarrow \tau\theta_k + (1 - \tau)\bar{\theta}_k$, $k = 1, 2$
18:      Update target policy network with $\bar{\phi} \leftarrow \tau\phi + (1 - \tau)\bar{\phi}$
19: **end for**

---

Table 1: MQES parameters

| | **Parameter** | | **Value** |
|---|---|---|---|
| Training | discount | | 0.99 |
| | Target smoothing coefficient | $\tau$ | 5e-3 |
| | Learning rate | | 3e-4 |
| | Optimizer | | Adam |
| | Replay buffer size | | 10e6 |
| | Batch size | | 256 |
| | Quantiles number | | 20 |
| | Action range | | [-1, 1] |
| | Environment steps per epoch | | 100 |
| Exploration | Exploration ratio | $\sqrt{2\alpha}$ | 0.8 |
| | Uncertainty ratio | $\beta$ | 1.6 |
| | Normalization factor | $C$ | 0.5 |

## D   HYPER-PARAMETERS SETTING

The hyper-parameters in our experiment are guaranteed to be consistent, as shown in Tab. 1.

## E   THRESHOLD SETTINGS FOR SPARSE TASKS

As illustrated in Sec. 5.3, We set the threshold according to the statistical analysis of untrained interaction behavior, using 99.9 quantile value, as shown in Tab. 2.

Table 2: Threshold settings for sparse reward tasks

| Sparse tasks | Threshold |
|---|---|
| Sparse Ant-v2 | 0.13 |
| Sparse Hopper-v2 | 0.605 |
| Sparse HalfCheetah-v2 | 3.5 |
| Sparse Walker2d-v2 | 0.525 |
| Sparse Humanoid-v2 | 0.17 |
| Sparse HumanoidStandup | 0.18 |

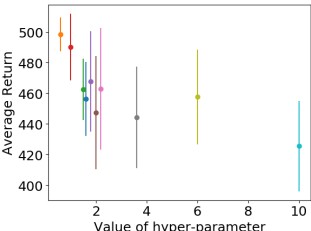

Figure 4: Ablation study on $\beta$

## F    ADDITIONAL EXPERIMENT RESULTS

Limited by the length of the text, we present the results of the evaluation on some simple tasks in Fig. 5. All evaluation data can be seen in Tab. 3.

The sensitivity to $\beta$ is shown in Fig. 4. $\beta$ controls the uncertainty magnitude, the smaller the value, the smaller the degree of optimism or pessimism, and vice versa. We evaluate the effect of $\beta$ in Ant-v2 task using $MQES\_Q$. What can be seen is that larger $\beta$ will degrade performance. Although smaller $\beta$ are more profitable in the Ant-v2 task, it is not necessarily in other task, so we set a uniform 1.6 in our experiments.

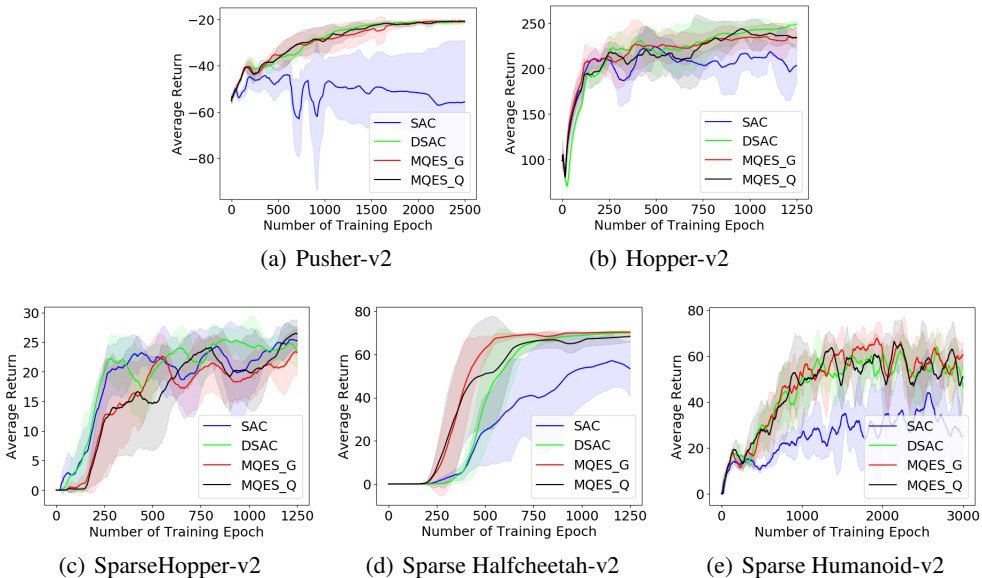

(a) Pusher-v2        (b) Hopper-v2

(c) SparseHopper-v2     (d) Sparse Halfcheetah-v2     (e) Sparse Humanoid-v2

Figure 5: Training curves on continuous control benchmarks in Mujoco. The x-axis indicates number of training epoch (100 environment steps for each training epoch), while the y-axis is the evaluation result represented by average episode return. The shaded region denotes one standard deviation of average evaluation over 5 seeds. Curves are smoothed uniformly for visual clarity.

Table 3: Average return over 5 seeds with one standard deviation at corresponding training step, i.e., 1.25 x $10^5$ million training step for Hopper-v2. The maximum value of each row is shown in bold.

| Task | 1e5 | SAC | DSAC | MQES_G | MQES_Q |
|---|---|---|---|---|---|
| Walker-v2 | 1.25 | $164.4 \pm 20.7$ | $204.2 \pm 15.3$ | $\mathbf{226.6} \pm 18.7$ | $212.1 \pm 18.7$ |
| Ant-v2 | 2.5 | $276.0 \pm 24.5$ | $456.5 \pm 52.2$ | $\mathbf{507.5} \pm 12.2$ | $456.5 \pm 48.4$ |
| Humanoid-v2 | 3.0 | $487.5 \pm 6.0$ | $597.7 \pm 4.4$ | $\mathbf{612.3} \pm 10.5$ | $607.3 \pm 13.6$ |
| Sparse Walker2d-v2 | 1.25 | $35.6 \pm 8.8$ | $38.8 \pm 4.9$ | $\mathbf{50.6} \pm 3.1$ | $47.9 \pm 3.7$ |
| Sparse Ant-v2 | 2.5 | $47.2 \pm 6.0$ | $60.3 \pm 8.4$ | $66.4 \pm 7.4$ | $\mathbf{69.3} \pm 15.3$ |
| Sparse HumanoidStandup | 0.25 | $5.3 \pm 10.2$ | $82.2 \pm 17.9$ | $\mathbf{95.0} \pm 0.8$ | $94.9 \pm 1.4$ |
| Pusher-v2 | 2.5 | $-55.7 \pm 26.9$ | $-21.1 \pm 0.6$ | $-21.0 \pm 0.6$ | $\mathbf{-20.8} \pm 0.3$ |
| Hopper-v2 | 1.25 | $203.3 \pm 19.9$ | $\mathbf{248.9} \pm 4.6$ | $233.6 \pm 7.6$ | $234.2 \pm 11.6$ |
| Sparse Hopper-v2 | 1.25 | $25.3 \pm 2.5$ | $23.7 \pm 2.9$ | $23.3 \pm 4.0$ | $\mathbf{26.4} \pm 2.3$ |
| Sparse HalfCheetah-v2 | 1.25 | $53.5 \pm 12.7$ | $70.2 \pm 1.1$ | $\mathbf{70.3} \pm 0.5$ | $68.3 \pm 2.4$ |
| Sparse Humanoid-v2 | 3.0 | $24.9 \pm 3.1$ | $46.6 \pm 8.6$ | $\mathbf{60.4} \pm 6.0$ | $51.0 \pm 10.3$ |

