# OpenReview forum: "MQES: Max-Q Entropy Search for Efficient Exploration in Continuous Reinforcement Learning"
_ICLR.cc/2021/Conference — Reject_

### Official Review · AnonReviewer4 · 2020-10-27

**Rating:** 4
**Confidence:** 3

**Review:**

The paper proposes an information-theoretic approach to exploration in model-free RL, by encouraging an exploration policy that is maximally informative about the optimal (distributional) value function. The authors discuss tractable approximation to this objective which can be implemented in continuous MDPs. The method is evaluated on benchmark continuous control tasks (Mujoco).

The basic underlying idea is interesting and, to the best of my knowledge, novel. However there are some major issues and/or limitations which should be addressed prior to publication.

**Clarity**
The paper is hard to follow and understand.
* There is a use of terminology which might not be clear or known for the general RL/exploration audience ("acquisition functions", "heteroscedastic aleatoric uncertainty"). The entire discussion of the two types of uncertainty seems somewhat disconnected from the method itself. A short explanation of why and how the two types of uncertainty are important *for the exploration problem* would be very helpful.
* There are some notation obscurities or inaccuracies. This is most notable in Section 4.1, which is unfortunate since this is where the key ideas of the approach are discussed.
    * Equation 8, which is central to what follows, is rather confusing. The text mentions that "$\pi_E$ selects action $a_t$ that (...)", but the equations then seems to define an entire policy. And the optimization problem (in the same Eq.) is in itself dependent on $a_t$, so it's not even clear one gets a valid policy/distribution from something like $\pi_E = \arg\max_\pi F^\pi(s_t,a_t)$.
    * Following the previous point, Eq. 9 is also confusing. It's not clear to me what the authors mean by measuring MI between $Z^*$ and *the pair* $(Z^\pi, \pi)$? It's also not clear what is the meaning of the "posterior distribution" denoted by $p$, and why the mutual information is measured for the Z parameters (return probs) but then re-expressed as the difference in entropies for the policies (action probs).

**Quality**
The paper has a good balance of a theoretically motivated algorithm, a practical implementation of it, and some basic empirical evaluation. Other than clarity issues discussed before, I have some concerns regarding the evaluation, and one more conceptual concern regarding the general idea:
* Since exploration here is encouraged by choosing informative actions about Q*, it's not clear that this method will be helpful in very sparse-reward settings (which are a central motivation for sophisticated exploration techniques). Put differently, relying on Q* to guide exploration ultimately couples exploration to the external reward, which seems rather undesirable to me. The method might be helpful from other perspectives (optimization, controlling the level of "over-exploration" etc), but it's not clear that it is helpful as an exploration method per se.
* The empirical evaluation is done only against rather limited baselines (basically random exploration baselines). I would encourage the authors to compare their method to other forms of exploration. Particularly relevant to this paper is the work by Houthooft et al. 2016 (VIME) which also uses information-theoretic objective for exploration in continuous problems. The authors should at least cite this paper in their related work section.
* Following the last two points, a great improvement could be if other than just demonstrating performance in terms of reward, the authors would evaluate the exploratory behavior itself of an agent trained with their method, in some simple environment (i.e in terms of novel states visited / distance traveled / etc.)

There are some more minor issues which should be addressed as well:
* In the abstract: the use of "optimism in the face of uncertainty" is definitely not a "recently"
* "The above methods are not efficient" (3rd paragraph, Introduction): This is not accurate. UCB (and other methods) **are** provably efficient for several problems/assumptions.
* Some relevant literature is missing from the related work. Most notable is the VIME paper mentioned earlier (Houthooft et al. NeurIPS 2016) and the Fox et al. 2018 ICLR paper (DORA The explorer) which combines counter-based like exploration with an optimism principle for high-dim MDPs.
* Section 3.2: $T^\pi$ is **not** the "Bellman optimiality operator" but rather the bellman operator for policy $\pi$.

**Conclusions**
This work has some interesting idea which could be useful for training RL agents in continuous problems. However in the current form of the paper it's hard to evaluate and understand some of the key ideas of the work. Given this, and together with the more conceptual concerns regarding evaluation and the basic approach, I think the paper is not ready for publication.

---

> ### Author Response · Authors · 2020-11-22
> **Response**
>
> 【Q1： There is a use of terminology which might not be clear or known for the general RL/exploration audience ("acquisition functions", "heteroscedastic aleatoric uncertainty"). The entire discussion of the two types of uncertainty seems somewhat disconnected from the method itself. A short explanation of why and how the two types of uncertainty are important for the exploration problem would be very helpful.】
>
> The introduction of "acquisition functions" is not related our main point of paper, hence we remove it in the revised version. Moreover, we describe the ”heteroscedastic aleatoric uncertainty" more detailed in the third paragraph in Sec. 1
>
> We add a short explanation of the benefits of uncertainty for RL exploration in the second paragraph at Sec. 1.
>
> 【Q2：  There are some notation obscurities or inaccuracies. This is most notable in Section 4.1, which is unfortunate since this is where the key ideas of the approach are discussed; Equation 8, which is central to what follows, is rather confusing. The text mentions that $\pi_E$ selects action $a_t$ that (...)", but the equations then seems to define an entire policy. And the optimization problem (in the same Eq.) is in itself dependent on $a_t$, so it's not even clear one gets a valid policy/distribution from something like $\pi_E(a_t|s_t) = \arg\max_{\pi} {\bf{F}} ^\pi(s_t, a_t)$; Following the previous point, Eq. 9 is also confusing. It's not clear to me what the authors mean by measuring MI between $Z^*$ and the pair  $(Z^\pi,\pi)$ ? It's also not clear what is the meaning of the "posterior distribution" denoted by $p$ , and why the mutual information is measured for the $Z$ parameters (return probs) but then re-expressed as the difference in entropies for the policies (action probs).】
>
> We admit that the theory presented in Sec. 4.1 is vague. In the revised version, we rewrite the equations to make them more accurate:
>
> For equation 8, the mutual information is conditioned on the state, Hence, to make it more rigorous, equation 8 is rewritten as: $\pi_E = \arg\max_{\pi\in \Pi} {\bf{F}} ^\pi(s_t).$
>
> For MQES, we consider actions as random variables and policies are the distributions that the actions follow. Hence, given state, the conditional mutual information is actually between the exploration action random variable $A_E\sim\pi_E$ and random variable of globally optimal $Q$ function  $Z^*(s,a^*)$ in Eq. 9, which are with value $a\in\bf{A}$ and $z^*(s,a^*)$, respectively. And the posterior probability $p(a|z^*(s,a^*),s)$ describes the distribution of exploration action conditioning on the state and globally optimal $Q$ function $z^*(s,a^*)$. Hence, Eq. 9 is rewritten as: $ {\bf{F}}^\pi(s_t) = {\bf{MI}}(Z^*(s,a^*),A|s = s_t) ={\bf{H}}\left[ \pi(a_t|s_t)\right]-{\bf{H}}\left[ p(a_t|z^*(s_t,a^*), s_t)\right].$
>
>  【Q3：Since exploration here is encouraged by choosing informative actions about Q*, it's not clear that this method will be helpful in very sparse-reward settings (which are a central motivation for sophisticated exploration techniques). Put differently, relying on Q* to guide exploration ultimately couples exploration to the external reward, which seems rather undesirable to me. The method might be helpful from other perspectives (optimization, controlling the level of "over-exploration" etc), but it's not clear that it is helpful as an exploration method per se.】
>
> Actually, epistemic uncertainty introduced by the estimation of $Q^*$ could offer extra information for exploration when external reward is sparse. If the states that are seldom visited, the epistemic uncertainty at those states will be relatively large and the exploration should be encouraged. Hence, it is expected to perform better at the sparse environments, and we also conduct the experiments at sparse mujoco environments to show the improvements.
>
> However, if we only formulate the uncertainty using ensemble critics, the formulated uncertainty is the mixture of the aleatoric and epistemic uncertainty, where the aleatoric uncertainty is caused by the randomness of the environment and cannot be eliminated. Hence, if we do not distinguish these two uncertainties and formulate them separately, we may explore the states visited frequently but with high randomness, i.e., low epistemic uncertainty and high aleatoric uncertainty, which is undesirable.

---

> > ### Author Response · Authors · 2020-11-22
> > **【Continued】Response**
> >
> > 【Q4：The empirical evaluation is done only against rather limited baselines (basically random exploration baselines). I would encourage the authors to compare their method to other forms of exploration. Particularly relevant to this paper is the work by Houthooft et al. 2016 (VIME) which also uses information-theoretic objective for exploration in continuous problems. The authors should at least cite this paper in their related work section.】
> >
> > Actually, VIME is different from our method. In VIME, the environment dynamics is estimated with stochastic parameters, and derive intrinsic reward sequentially. It is more like model-based method. However, in MQES, we model the uncertainty by using distributional and ensemble Q function approximations. So, MQES is a model-free method.
> >
> > 【Q5：Following the last two points, a great improvement could be if other than just demonstrating performance in terms of reward, the authors would evaluate the exploratory behavior itself of an agent trained with their method, in some simple environment (i.e in terms of novel states visited / distance traveled / etc.)】
> >
> > Demonstrating the insights of MQES in the simple environments is definitely a good way to improve the quality of our method. But due to the time limit of the rebuttal, we will put it as future work.
> >
> > 【Q6：In the abstract: the use of "optimism in the face of uncertainty" is definitely not a "recently"】
> >
> > We have removed "recently" in the revised version.
> >
> > 【Q7："The above methods are not efficient" (3rd paragraph, Introduction): This is not accurate. UCB (and other methods) are provably efficient for several problems/assumptions.】
> >
> > Please see the first part of this sentence, i.e., "However, since the aleatoric uncertainty in the RL systems are heteroscedastic, ...". We point out that UCB could be not efficient when the aleatoric uncertain
> >
> > 【Q8： Some relevant literature is missing from the related work. Most notable is the VIME paper mentioned earlier (Houthooft et al. NeurIPS 2016) and the Fox et al. 2018 ICLR paper (DORA The explorer) which combines counter-based like exploration with an optimism principle for high-dim MDPs.
> > 】
> >
> > Since VIME and DORA are all intrinsic motivated exploration method, We refer the in the first paragraph in the Sec. 2.
> >
> > 【Q9：Section 3.2: $T^\pi$ is not the "Bellman optimiality operator" but rather the bellman operator for policy $\pi$.】
> >
> > We have corrected "Bellman optimality operator" as "Bellman operator" in the Section 3.

---

### Official Review · AnonReviewer2 · 2020-10-28
**Are the empirical results significant enough to support the proposed exploration heuristic?**

**Rating:** 3
**Confidence:** 4

**Review:**

The paper proposes an exploration scheme for RL in continuous action spaces using the principle of information maximization for globally optimal Q distribution.

1. I felt that the paper isn't well written and discusses a lot of different concepts in a haphazard manner.  There are a lot of equations and symbols in the text without proper explanation and context which make it difficult to gather the main contribution. The language used gets vague in many statements made in the paper. For ex. "Proposition 1. Generally, the posterior probability is as follows".

2. A lot of algorithm adaptations are proposed without actually carrying out ablations which make it difficult to discern if the proposed MI maximization is indeed responsible for performance. For ex. "Since the target for critic in the advanced algorithms, like SAC and TD3, is usually estimated pessimistically..". The authors should actually present ablations to support if a pessimistic estimate is indeed required for their adaptation for these methods.

3. Why haven't the authors included OAC as a baseline given that it outperforms SAC in several tasks? Further the results show little difference in performance in comparison with DSAC on the mujoco tasks, given that only 5 seeds were used in evaluation, it brings the significance of the results under question. The authors should provide appropriate measures like P-values to support the experiments.

---

> ### Author Response · Authors · 2020-11-22
> **Response**
>
> 【Q1：I felt that the paper isn't well written and discusses a lot of different concepts in a haphazard manner. There are a lot of equations and symbols in the text without proper explanation and context which make it difficult to gather the main contribution. The language used gets vague in many statements made in the paper
> For ex. "Proposition 1. Generally, the posterior probability is as follows".】
>
> We admit that we bring a few concepts that are not generally known to RL researchers. To make the background easy to follow, we firstly explain the motivation of epistemic and aleatoric uncertainty encouraging exploration in Sec. 1; Then, we remove the introduction of acquisition function, which is not our main point.
>
> For the equations, we rewrite Sec. 4.1 to make the derivation of MQES more rigorous, which is the main theoretical contribution of our paper; Then, we check all the symbols and try to clarify them.
>
>
> For the language, we have polished the paper.
>
> 【Q2：A lot of algorithm adaptations are proposed without actually carrying out ablations which make it difficult to discern if the proposed MI maximization is indeed responsible for performance. For ex. "Since the target for critic in the advanced algorithms, like SAC and TD3, is usually estimated pessimistically..". The authors should actually present ablations to support if a pessimistic estimate is indeed required for their adaptation for these methods.】
>
> We admit that we need more ablation experiments. In the revised version, we added Sec. 5.4 to conduct ablation experiments, regarding to sesitivity to the hyper-parameters and the gain of distinguishing two types of uncertainty.
>
> However, it is worth noting that we can utilize other methods to formulate $Z^{\pi_E}$, like mean estimation, i.e., $\mathbb{E} \left[Z^{\pi_E}\right]=\mu_Z (s, a;\theta)$ and $z_i^{\pi_E}(s,a;\theta) = \mathbb{E}_{k = {1, 2}} \left[z_i(s, a; \theta_k)\right]$. But, it only affects the choice of hyper-parameter $\beta$ and do not affect the final performance.
>
> 【Q3： Why haven't the authors included OAC as a baseline given that it outperforms SAC in several tasks? Further the results show little difference in performance in comparison with DSAC on the Mujoco tasks, given that only 5 seeds were used in evaluation, it brings the significance of the results under question. The authors should provide appropriate measures like P-values to support the experiments.】
>
> We compare with OAC in Sec. 5.4.1, whose performance is similiar with DSAC, since it cannot avoid the effects of aleatoric uncertainty. Besides, we add more experiments on sparse Mujoco tasks and the results are discussed more detailed in Sec. 5. In standard Mujoco, MQES does perform slightly better than DSAC in those easy tasks such as Hopper-v2 and Walker2D-v2. However, in those hard and sparse-reward tasks, MQES performance significantly better than DSAC, and MQES\_Q demonstrates the advantages of stability than MQES\_G.

---

### Official Review · AnonReviewer1 · 2020-10-28

**Rating:** 5
**Confidence:** 4

**Review:**

---
Summary

This paper studies the problem of efficient exploration in continuous environment. It proposes a novel algorithm, Max-Q Entropy Search (MQES), and utilizes Distributional SAC (DSAC) to formulate the uncertainties. Experiments show that the proposed algorithm, MQES, outperforms the baselines (SAC, DSAC).

---
Comments

However, MQES doesn't show significant improvement over DSAC. For example, Except Sparse-HalfCheetah-v2, MQES_Q has almost same performance as DSAC. Except Sparse-HalfCheetah-v2 and Ant-v2, MQES_G has almost the same performance as DSAC.

Another question is: The horizon is cut to 100 while most papers and OpenAI Gym use 1000 by default. Why do the authors choose 100? How does MQES perform with longer horizon, like 1000?

The paper shows that exploring using both aleatoric and epistemic uncertainty can improve the performance. What if we consider only one of them, e.g., using only aleatoric uncertainty? I'd like to see this as ablation.


---
Writing Quality

The writing can also be improved.

Table 1: Could you please highlight all algorithms within 1 std to the best?

Sparse-HalfCheetah-v2: Could you please provide more details about the environment?

What does the mutual information between $(Z^*, \pi^*)$ and $(Z^{\pi_E}, \pi_E)$ mean? Are $\pi^*$ and $\pi_E$ random variables? Moreover, in deterministic environments (as in Mujoco environments), $\pi^*$ is also deterministic, so is $Z^*(s_t, a_t)$.

Eq 8: LHS is a scalar (probablity), while RHS is a policy. Please clarify notations to avoid confusion.

Eq 9: What is $p$? The first input of MI is simply a $Z^*$ while the second is a pair $(Z^\pi, \pi(a_t | s_t))$. Please elaborate.

> To measure the intractable distribution of $Z^*$ during training, we use the $\hat Z^*$ for approximation

Please rephrase it and say that $\hat Z^*$ will be defined later.

Eq 20: This is not an unbiased estimation, as $\mathbb{E}[X^{-1}] \neq \mathbb{E}[X]^{-1}$. Also please clarify K.

---

> ### Author Response · Authors · 2020-11-22
> **Response**
>
> 【Q1: However, MQES doesn't show significant improvement over DSAC. For example, Except Sparse-HalfCheetah-v2, $MQES_Q$ has almost same performance as DSAC. Except Sparse-HalfCheetah-v2 and Ant-v2, $MQES_G$ has almost the same performance as DSAC. 】
>
> As shown in Figure 1, both MQES\_G and MQES\_Q show better performance than DSAC in the difficult tasks, and in other easy tasks as shown in Appendix F, MQES can also show slightly better than DSAC. It seems that exploration is not the main bottleneck in the easy tasks, i.e., standard Hopper-v2 and Walker2D-v2. However, in those harder or sparse reward tasks, MQES performs significantly better than DSAC, and MQES\_Q demonstrates the advantages of stability than MQES\_G.
>
> 【Q2: Another question is: The horizon is cut to 100 while most papers and OpenAI Gym use 1000 by default. Why do the authors choose 100? How does MQES perform with longer horizon, like 1000? 】
>
> One consideration for our shortened episode length is training efficiency. Also, if the environment, sampling and training settings are the same as baseline, such comparison is fair, so we believe that the horizon doesn't matter as long as it isn't extremely outrageous.
>
> 【Q3: The paper shows that exploring using both aleatoric and epistemic uncertainty can improve the performance. What if we consider only one of them, e.g., using only aleatoric uncertainty? I'd like to see this as ablation.】
>
> In Sec 5.4.1, we provide an ablation study to show the performance gain brought by distinguishing the two types of uncertainty.
>
> 【Q4:  Table 1: Could you please highlight all algorithms within 1 std to the best?】
>
> In the revised version, we highlight the mean to the best, which is more appropriate
>
> 【Q5: Sparse-HalfCheetah-v2: Could you please provide more details about the environment?】
>
> We have added experiments on more sparse tasks, which is shown in Sec. 5.3. We describe detailed settings for sparse reward and show that MQES can show stable and consistent advantage over DSAC in those sparse tasks. Briefly, standard Mujoco gives precise reward each step, while in sparse setting, the reward is given only when the agent moves through the threshold (see section 5.3 for more details.).
>
> 【Q6： What does the mutual information between $(Z^*,\pi^*)$  and $(Z^{\pi_E},\pi_E)$ mean? Are $\pi^*$ and $\pi_E$ random variables? Moreover, in deterministic environments (as in Mujoco environments), $Z^*$ is also deterministic, so is $\pi^*$; What is $p$? The first input of MI is simply a  while the second is a pair . Please elaborate.】
>
> In the revised version, we have rewritten theoretical part, i.e., Section 4.1. Specifically, for MQES, we consider actions as random variables and policies are the distributions that the actions follow. Hence, given state, the conditional mutual information is actually between the exploration action random variable $A_E\sim\pi_E$ and random variable of globally optimal $Q$ function  $Z^*(s,a^*)$ in Eq. 9, which are with value $a\in\bf{A}$ and $z^*(s,a^*)$, respectively. And the posterior probability $p(a|z^*(s,a^*),s)$ describes the distribution of exploration action conditioning on the state and globally optimal $Q$ function $z^*(s,a^*)$. Hence, Eq. 9 is rewritten as: $ {\bf{F}}^\pi(s_t) = {\bf{MI}}(Z^*(s,a^*),A|s = s_t) ={\bf{H}}\left[ \pi(a_t|s_t)\right]-{\bf{H}}\left[ p(a_t|z^*(s_t,a^*), s_t)\right].$
>
> 【Q7： What does the mutual information between $(Z^*,\pi^*)$  and $(Z^{\pi_E},\pi_E)$ mean? Are $\pi^*$ and $\pi_E$ random variables? Moreover, in deterministic environments (as in Mujoco environments), $Z^*$ is also deterministic, so is $\pi^*$.】
>
> For Eq. 8, the mutual information is conditioned on the state, which is the expectation of the exploration policy. Hence, to make it more rigorous, Eq. 8 is corrected as $\pi_E = \arg\max_{\pi\in \Pi} {\bf{F}} ^\pi(s_t).$
>
> 【Q8：To measure the intractable distribution  of $Z^*$  during training, we use the  $\hat{Z}^*$  for approximation Please rephrase it and say that $\hat{Z}^*$  will be defined later； This is not an unbiased estimation and also please clarify K】
>
> We have corrected the corresponding part.

---

### Official Review · AnonReviewer3 · 2020-10-29
**MaxQ Entropy Search Blind Review #1**

**Rating:** 6
**Confidence:** 3

**Review:**

This work introduces max-Q Entropy Search (MQES) exploration principle  for continuous RL algorithms. MQES addresses the exploration-exploitation dilemma that constitutes a fundamental RL problem. Actually, MQES defines an exploration policy able to explore optimistically and avoid over-exploration. One of the main advantages of MQES is its ability to recognise the epistemic and aleatoric uncertainty. Empirical analysis has been conducted on Mujoco, showing that the performance of MQES is comparable to  those of other state-of-the-art algorithms.

In general the paper is well written and can be easily followed by the reader. Nevertheless some parts of the MQES should be explained in more detail. For instance, authors should give more details about the target policy introduced at Section 4.3. Actually, the reader should check Algorithm 2 at Appendix in order to understand its purpose and how the target policy is updated. I think that it would be better Algorithm 2 to be moved in the main paper if it is possible. Another point that should be discussed more clearly is the impact of the 3 hyper-parameters (\alpha, \beta, and C) on the performance of MQES. To be more specific, why did you set the uncertainty ratio equal to 1.6? Finally, the empirical results are not discussed at all. It seems for example that the performance of MQES_G is more stable compared to that of MQES_Q. Moreover, the performance of DSAC is almost equal (or better) to that of MQES_Q. All these points should be explained or discussed by the authors.

---

> ### Author Response · Authors · 2020-11-22
> **Response**
>
> 【Q1: authors should give more details about the target policy introduced at Section 4.3. Actually, the reader should check Algorithm 2 at Appendix in order to understand its purpose and how the target policy is updated. I think that it would be better Algorithm 2 to be moved in the main paper if it is possible】
>
> We agree that it is definitely better to move Algorithm 2 to the main paper However, due to the paper length limit, maybe we could not move it now。
>
> 【Q2: Another point that should be discussed more clearly is the impact of the 3 hyper-parameters ($\alpha$, $\beta$, and $C$) on the performance of MQES. To be more specific, why did you set the uncertainty ratio equal to 1.6?】
>
> To clarify how the hpyer-parameters affect the performance, we conduct ablation study on hyper-parameters and results are shown in Sec. 5.4.2 and appendix F, and discuss the impact of the hyper-parameters on the performance accordingly.
>
> 【Q3:  Finally, the empirical results are not discussed at all. It seems for example that the performance of $MQES_G$ is more stable compared to that of $MQES_Q$. Moreover, the performance of DSAC is almost equal (or better) to that of $MQES_Q$. All these points should be explained or discussed by the authors.】
>
> In the standard mujoco tasks and the reward is dense,  MQES\_G is more stable compared to that of MQES\_Q in the easy tasks, e.g., Hopper-v2 and Walker-v2, but in the difficult tasks such as Ant-v2 and the sparse reward mujoco tasks (the tasks shown in Sec 5.3), MQES\_Q is shown to be more stable. It is mainly because the policy follows Gaussian distribution, which renders the value function random variable more Gaussian in the easy tasks due to the relatively low action and state space dimension. Hence, in the easy tasks, the Gaussian formulation incorporates this prior into the learning. Hence, the MQES\_G should be more stable than MQES\_Q in the easy tasks.
>
> However, in the difficult tasks, which are with relatively higher state and action space or sparse (or un-smooth) reward, the Gaussian assumptions will not hold, and the quantile formulation should be more reasonably since it can represent distribution in the more flexible way.\par
>
> As shown in Figure 1, both MQES\_G and MQES\_Q show better performance than DSAC in the difficult tasks, and in other easy tasks as shown in Appendix F, MQES can also show slightly better than DSAC. It seems that exploration is not the main bottleneck in the easy tasks, i.e., standard Hopper-v2 and Walker2D-v2. However, in those harder and sparse reward tasks, MQES performs significantly better than DSAC, and MQES\_Q demonstrates the advantages of stability than MQES\_G.

---

### Official Review · AnonReviewer5 · 2020-11-07
**Interesting idea but unconvincing results.**

**Rating:** 4
**Confidence:** 3

**Review:**


This paper proposes MQES, a Max-Q entropy search for policy optimization in continuous RL. The authors propose to combine advantages of the information-theoretic principle and distributional RL, in which epistemic and aleatoric uncertainty are estimated using similar entropy-search acquisition functions in the Bayesian Optimization (BO). As said, this is a new method to introduce a more efficient exploration strategy. As a result, policy improvement is formulated as a constraint optimization problem where a next exploration policy can be solved in a closed-form. The proposed method is evaluated on Mujoco tasks and compared against other off-policy approaches, SAC, and DSAC. The results show MQES outperforms other methods in domains where exploration is needed.

The main contribution of the paper is to introduce an approximation to distribution Q functions that are based on the epistemic and aleatoric uncertainty. The main objective is based on the mutual information maximization as described in 4.1. A practical implementation is proposed in 4.1. The idea makes sense however the presentation and its experiment results make it hard to understand some important details. Some of my major comments are as follows.

1. Although the idea is interesting, it's not yet clear how the proposed method can be used as an additional module on top of other entropy-regularized off-policy approaches like SAC or TD3, etc. The paper can benefit more if it can be formulated in such a more general way.

2. The practical implementation seems to make sense. However, it's still unclear to me how policies \pi_E and \pi_T are parameterized. Although a sketch of the main idea is described in Algorithm 1, it's not clear to me each term is parameterized and computed based on a particular parameterization.


3. An updated policy as a solution of (18) gives an update on the mean but keeps the covariance unchanged. I was wondering then how this policy can adaptively change its exploration through the progress of learning?

4. The experiment results are quite preliminary. There are no experiment settings. There are a number of hyperparameters of MQES that might affect overall performance of MQES, e.g. N, \beta etc. but not discussed and ablated? The comparisons might also take into account other distributional policy search methods.

5. And some minor comments

	- Eq. 4 and 5, 6: min and log instead of arg functions?

	- Eq. 9: What is the difference between posterior p(a|Z^*(s,\pi^*) and \pi^*? Is the posterior not the optimal policy since Z^* is the optimal distributional value function estimate? A detailed derivation for Eq.9 is expected.

	- Definitions for terms in 4.1:  p(a|Z^*(s,\pi^*) vs  p(a|Z^*(s,\pi) p(a|Z^*(s,\pi^*) p(a|Z^*(s,\pi) etc.

	- in Eq.14: Should the aleatoric uncertainty be the variance of the {min_{\theta_k} z_i(\theta_k)} instead of en expectation over \theta_k. Because z_i is estimated as the min of two estimates, e.g. in Eq.4

	- Some theoretical steps are not clearly justified of why.

	- is $n$ used in Proposition 2?

	- the conventions of CDF in Proposition 2 and in Eq.21 should be made consistent with its first definition in Eq.11

	- definition of G() is not used after its first definition.

	- why the horizon is set to 100, instead of 1000 environment steps like in the SAC paper?

---

> ### Author Response · Authors · 2020-11-22
> **Response**
>
>
> 【Q1: Although the idea is interesting, it's not yet clear how the proposed method can be used as an additional module on top of other entropy-regularized off-policy approaches like SAC or TD3, etc. The paper can benefit more if it can be formulated in such a more general way.】
>
> Actually, MQES is indeed a generally exploration framework for off-policy actor-critic algorithms. Nevertheless, as an example, we incorporate MQES-based exploration into SAC in Section 4.2. Furthermore, if the policy is deterministic (like TD3), the exploration policy is just a special case of Eq. 13, which is without covariance matrix.
>
> 【Q2: The practical implementation seems to make sense. However, it's still unclear to me how policies $\pi_E$ and $\pi_T$ are parameterized. Although a sketch of the main idea is described in Algorithm 1, it's not clear to me each term is parameterized and computed based on a particular parameterization.】
>
> In Algorithm 1, we show that the target policy $\pi_T$ is parameterized by $\phi$. And we use Eq. 13 to derive $\pi_E$, which introduces no extra parameters.
>
> 【Q3: An updated policy as a solution of (18) gives an update on the mean but keeps the covariance unchanged. I was wondering then how this policy can adaptively change its exploration through the progress of learning?】
>
> In the proof of Proposition 2, we prove that the covariance matrices of target and exploration policy are equal. Please check Appendix B for more details.
>
> 【Q4: The experiment results are quite preliminary. There are no experiment settings. There are a number of hyper-parameters of MQES that might affect overall performance of MQES, e.g. N, $\beta$ etc. but not discussed and ablated? The comparisons might also take into account other distributional policy search methods】
>
> In the revised paper, we explain the experiment settings in more detail in Sec. 5.1.
>
> Also, we conduct ablation study on hyper-parameters and results are shown in Sec. 5.4.2 and appendix F, and discuss the impacts of the hyper-parameters on the performance.
>
> For the benchmarks, as mentioned in Sec. 5.1, DSAC has compared with TD4 (the distributional version of TD3), and outperformed TD4 in most of tasks. So there's no need comparing with TD4, and we choose to implement our MQES based on DSAC and also compared with it, which obtains good results.
>
> 【Q5: Eq. 4 and 5, 6: min and log instead of arg functions? 】
>
> Sorry for our negligence, and we have corrected the typo.
>
> 【Q6: Eq. 9: What is the difference between posterior $p(a|Z^*(s,\pi^*)$ and $\pi^*$? Is the posterior not the optimal policy since $Z^*$ is the optimal distributional value function estimate? A detailed derivation for Eq.9 is expected; Definitions for terms in 4.1: $p(a|Z^*(s,\pi^*)$ vs $p(a|Z^*(s,\pi) p(a|Z^*(s,\pi^*) p(a|Z^*(s,\pi)$ etc; Some theoretical steps are not clearly justified of why.】
>
> We admit that the theory presented in Sec. 4.1 is vague. In the revised version, we rewrite the equations in Sec. 4.1 to make them more accurate:
>
>  1.  For equation 8, the mutual information is actually conditioned on the state. Hence, to make it more rigorous, equation 8 is rewritten as: $\pi_E = \arg\max_{\pi\in \Pi} {\bf{F}} ^\pi(s_t).$
>
> 2. For MQES, we consider actions as random variables and policies are the distributions that the actions follow. Hence, given state, the conditional mutual information is actually between the exploration action random variable $A_E\sim\pi_E$ and random variable of globally optimal $Q$ function  $Z^*(s,a^*)$ in Eq. 9, which are with value $a_E\in\bf{A}$ and $z^*(s,a^*)$, respectively. And the posterior probability $p(a|z^*(s,a^*),s)$ describes the distribution of exploration action conditioning on the state $s$ and value $z^*(s,a^*)$ of globally optimal $Q$ function random variable $Z^*(s,a^*)$. Hence, Eq. 9 is rewritten as: $ {\bf{F}}^\pi(s_t) = {\bf{MI}}(Z^*(s,a^*),A|s = s_t) ={\bf{H}}\left[ \pi(a_t|s_t)\right]-{\bf{H}}\left[ p(a_t|z^*(s_t,a^*), s_t)\right].$
>
>
> 【Q7: Should the aleatoric uncertainty be the variance of the ${min_{\theta_k} z_i(\theta_k)}$ instead of en expectation over $\theta_k$. Because $z_i$ is estimated as the min of two estimates, e.g. in Eq.4】
>
> The aleatoric uncertainty is a property of the environment itself, whereas $Z_{\pi}$ is the result of our pessimistic estimation, and the two are not directly related. At the same time, we need to estimate the aleatoric uncertainty of the environment realistically in order to accurately avoid it in our explorations, and it makes no sense to estimate aleatoric uncertainty.

---

> > ### Author Response · Authors · 2020-11-22
> > **【Continued】Response**
> >
> > 【Q8:  is $n$ used in Proposition 2?】
> >
> > In proposition 2, the length of vector $m$ is the action dimension $n$.
> >
> > 【Q9: the conventions of CDF in Proposition 2 and in Eq.21 should be made consistent with its first definition in Eq.11】
> >
> > We unify the conventions of CDF in the revised version.
> >
> > 【Q10: definition of $G(\cdot)$ is not used after its first definition.】
> >
> > In the revised version, we use $G(\cdot)$ in the Proposition 2.
> >
> > 【Q11: why the horizon is set to 100, instead of 1000 environment steps like in the SAC paper?】
> >
> > One consideration for our shortened episode length is training efficiency. Also, if the environment, sampling and training settings are the same as baseline, such comparison is fair, so we believe that the horizon doesn't matter as long as it isn't extremely outrageous.

---

### Author Response · Authors · 2020-11-22
**Thanks to all the reviewers. Summary of changes and paper revision.**

We greatly appreciate the reviews you provided on our paper. We are very pleased to get the valuable comments and excellent suggestions for further improving our work. Revisions have been made in the paper accordingly. The revisions are summarized as follows:

1.  For the theoretical part, i.e., Sec. 4.1, we rewrite the Eq. 8, 9 and 10 to
 make them more rigorous and show our theoretical contribution more clearly.


2.  For the results part, to make our results moreconvincing,  we conduct experiments in the sparse Mujoco environments, and describes more details of the setting. Also, we have the ablation study to show sensitivity of hyper-parameters and gain of distinguishing two types uncertainty.


3. Expect the theoretical and results part, to make our paper easier to follow, we polish our paper from the perspective of paper structure, symbol clarifications and language, according to the comments


A point-by-point comment-response section is given next.  Please note that all the equations and references are referred to those in the revised version of the manuscript unless otherwise indicated. Main changes in the revised manuscript are highlighted in red for the ease of cross references. Hope our responses would clarify your concerns. We are looking forward to your future comments if any.

---

### Decision · Program_Chairs · 2021-01-07
**Final Decision**

**Decision:**

Reject

**Comment:**

The paper contributes to the community by introducing an approximation to distribution Q functions, based on the epistemic and aleatoric uncertainty. The reviewers believe the ideas make sense. However the presentation and its experiment results make it hard for them to understand some important details. For example, the reviewers are confused about why the empirical results show the proposed methods are better.

The majority of the reviewers are negative about the paper. After rebuttal, the reviewers are not convinced. Based on this, the meta-reviewer recommends rejection. Authors can strengthen paper by improving its presentation and addressing the concerns from the reviewers.